# Relationship between Respiratory Rate, Oxygen Saturation, and Blood Test Results in Dogs with Chronic or Acute Respiratory Disease: A Retrospective Study

**DOI:** 10.3390/vetsci11010027

**Published:** 2024-01-10

**Authors:** Yuta Nakazawa, Takafumi Ohshima, Mami Kitagawa, Takaomi Nuruki, Aki Fujiwara-Igarashi

**Affiliations:** 1Laboratory of Veterinary Radiology, Department of Veterinary Medicine, School of Veterinary Medicine, Nippon Veterinary and Life Science University, 1-7-1 Kyonancho, Musashino, Tokyo 180-8602, Japan; yuta.veterinary@gmail.com (Y.N.); tatsu2627@outlook.jp (T.O.); 2TRVA Emergency Animal Medical Center, 8-19-12 Fukasawa, Setagaya, Tokyo 158-0081, Japan; vet_mamitus@yahoo.co.jp (M.K.); taka.vet.0323@gmail.com (T.N.)

**Keywords:** canine, respiratory disease, pulse oximetry, blood gas analysis, C-reactive protein

## Abstract

**Simple Summary:**

Diagnosis of respiratory diseases uses a variety of tools in addition to signalment and clinical signs, including physical examination, blood tests, and diagnostic imaging. Until now, studies have been conducted on the relationship between specific respiratory diseases and physical examination and blood test findings, but there have been no reports on these test findings for respiratory diseases as a whole. This study aimed to investigate the association of physical examinations and blood tests with canine respiratory diseases and to compare the examination findings in the chronic and acute phases. Significant associations were observed with various items mainly in lung diseases, such as increased respiratory rate, decreased oxygen saturation, and increased C-reactive protein in both chronic and acute phases. Blood gas analysis showed that respiratory alkalosis was most prevalent in the chronic phase, whereas respiratory acidosis was most prevalent in the acute phase. The results obtained in this study demonstrate that these testing tools are useful for diagnosis and monitoring or treatment responses in respiratory diseases.

**Abstract:**

This study aimed to investigate the association of respiratory rate (RR), oxygen saturation (SpO_2_), and blood findings with respiratory disease in dogs and to compare the examination findings in the chronic and acute phases. Dogs that visited a veterinary referral hospital with respiratory symptoms were classified into the chronic disease group (GC), and those that visited the emergency veterinary hospital were classified into the acute disease group (GA). In total, 704 and 682 dogs were included in GC and GA, respectively. The RR and SpO_2_ were significantly higher and lower, respectively, in patients with lung disease compared to other disease sites in both groups. White blood cell counts were significantly increased in patients with lung and pleural diseases in both groups. Respiratory alkalosis and respiratory acidosis were most common in GC and GA, respectively. The C-reactive protein levels were elevated in both groups, primarily in patients with lung disease. Associations between the results of several tests for understanding and diagnosing respiratory conditions and diseases were recognized, and differences in the trends of the chronic and acute phases were clarified. These tools may be used as adjuncts to other tests for the diagnosis and monitoring of treatment responses.

## 1. Introduction

In veterinary practice, respiratory diseases are diagnosed using various tools, including clinical signs, history, physical examination, blood tests, and imaging. Research has been conducted on the relationships and characteristics of various respiratory diseases using clinical signs and imaging. However, in emergency cases, patient stabilization may be prioritized before history taking or imaging; therefore, it is necessary to accurately understand respiratory conditions.

Physical examination is a quick and simple method for determining the respiratory condition of small animals and may help identify abnormalities in specific parts of the respiratory system. An increased respiratory rate (RR) is thought to indicate lung or pleural cavity disease [1]. However, few studies have investigated these associations, and one study found no significant association between an increased RR and lung or pleural disease [1].

Oxygen saturation (SpO_2_) can be measured using a pulse oximeter, which is a simple and less invasive test that assesses blood oxygenation through the skin without the need for arterial catheter placement. Owing to its convenience, SpO_2_ is widely used for continuous and long-term monitoring in both human and veterinary patients. In veterinary medicine, there have been studies on SpO_2_ fluctuations during anesthesia [2,3,4,5], and SpO_2_ is known to be affected by patient-related factors, such as motion artifacts, skin pigment, hypotension, and severe anemia [6,7,8]. The only study comparing SpO_2_ with the partial pressure of oxygen (PaO_2_) in the arterial blood, the gold standard for assessing oxygenation, reported that SpO_2_ is not a clinically relevant substitute for PaO_2_ [5]. Therefore, although a decreased SpO_2_ is associated with hypoxemia in clinical practice, SpO_2_ fluctuations in respiratory diseases have not been evaluated.

Blood test findings may also guide the diagnosis and treatment of respiratory symptoms in dogs. In particular, blood gas analysis provides rapid information on ventilatory and metabolic status for acid–base status assessment [9,10]. In respiratory medicine, arterial blood gas analysis is considered the gold standard for assessing both acid–base status and ventilatory oxygenation. However, various studies have reported that some parameters, including pH, bicarbonate (HCO_3_) concentration, and base excess (BE) in venous blood gas analysis, correlate with those in arterial blood [11,12]. Additional important blood tests included complete blood count (CBC) and C-reactive protein (CRP) levels. The complete blood count is one of the most commonly evaluated blood parameters in clinical practice and is useful in diagnosing various diseases, such as anemia, infectious diseases, and leukemia [13]. C-reactive protein levels are elevated in dogs with acute inflammatory and non-inflammatory conditions such as immune-mediated hemolytic anemia and thrombocytopenia [14,15]. Among respiratory diseases, CRP levels have primarily been studied in lung diseases and have been reported to be elevated in bacterial pneumonia, bronchopneumonia, and aspiration pneumonia [14,16,17].

Consequently, physical examinations, SpO_2_, and various blood tests are useful for dogs with respiratory symptoms because they are simple and quick to perform. Previously, characteristics of blood test findings have been described in specific respiratory diseases, such as elevated white blood cell and CRP levels in bacterial pneumonia and aspiration pneumonia [16,17,18,19]. However, no studies have compared and evaluated the usefulness of these items across respiratory diseases, and SpO_2_, in particular, has rarely been investigated. Therefore, the purpose of this study was to investigate the association between RR, SpO_2_, and various blood test findings in respiratory diseases and to compare these findings in chronic and acute diseases.

## 2. Materials and Methods

### 2.1. Study Design

A retrospective study was conducted using the medical records of dogs who visited the Department of Respiratory Medicine at the Veterinary Medical Teaching Hospital of Nippon Veterinary and Life Science University (VMTH-NVLU) between April 2017 and March 2022 and dogs who visited TRVA Emergency Animal Medical Center (TRVA) between July 2020 and March 2022. Among the dogs presenting with respiratory symptoms that had diagnosed with or suspected to have respiratory disease, the study included dogs with a clinical provisional or definitive diagnosis and documented the results of at least one of the five items (RR, SpO_2_, CBC, blood gas analysis, and CRP level). Of the dogs with respiratory signs, those diagnosed with non-respiratory diseases were excluded. Patients who visited the VMTH-NVLU were classified into the chronic respiratory disease group (Group C; GC), as all patients presented with clinical signs for more than 1 week. In contrast, patients who visited the TRVA were classified into the acute respiratory disease group (Group A; GA) because all patients had clinical signs for less than 1 week. Group C was subclassified into the following six categories based on the site of involvement: nasal cavity, pharynx, larynx, trachea/bronchi, lung, and mediastinum or pleural cavity. In GA, it was difficult to distinguish between pharyngeal and laryngeal diseases due to the acute nature of the disease, and the diagnosis was divided into the following five categories: nasal cavity, laryngopharynx, trachea/bronchi, lung, and mediastinum or pleural cavity. Duplicate, triplicate, and quadruplicate cases were included when a dog was found to have multiple diseases.

### 2.2. Diagnosis

Diagnosis of respiratory diseases was performed by two veterinarians specializing in respiratory medicine at VMTH-NVLU and two veterinarians specializing in emergency medicine at TRVA. Signalment, history taking, and physical examinations were initially performed in all patients for the diagnosis of respiratory diseases. Blood tests were performed based on the patient’s symptoms and condition. Imaging tests included radiography, fluoroscopy, and ultrasonography, depending on the patient’s symptoms and suspected disease site. However, if the owner did not wish for the patient to undergo further examinations or histological diagnosis under general anesthesia or sedation because of anesthetic risk or costs; if the examination could not be performed due to worsening respiratory conditions; or if the patient died, a provisional diagnosis was made based on radiographic, ultrasonography, and therapeutic response findings.

### 2.3. RR, SpO_2_, and Various Blood Tests

The investigation parameters and reference values for the RR, SpO_2_, CBC, blood gas, and CRP levels used in this study are listed in Appendix A. Respiratory rate and SpO_2_ measurements were taken at the first visit, and blood tests were taken at the time of first visit. The number of RR per minute was measured. Pulse oximetry (New Radical-7, Masimo, Discovery, Irvine, CA, USA) was performed on the lip, ear, and tail, depending on the case. In addition, the values measured under oxygen inhalation were not included. Blood was obtained from the jugular, cephalic, or lateral saphenous veins. For blood gas analysis, blood was collected from the femoral or dorsalis pedis arteries in some patients. The survey items for CBC (Celltac α, NIHON KOHDEN, Nishiochiai, Shinjuku-ku, Tokyo, Japan) were as follows: red blood cell (RBC) count, hematocrit (HCT), hemoglobin levels (HGB), white blood cell (WBC) count, and platelet count. In the WBC count, neutrophils (band or segmented), lymphocytes, monocytes, and eosinophils were investigated. Blood gas analysis (GEM PREMIER 3500, IL Japan, Mita, Minato-ku, Tokyo, Japan) was performed using pH, partial pressure of oxygen and carbon dioxide (PCO_2_), HCO_3_ concentration, and BE. Acid–base disorders in the venous and arterial blood gas analyses were evaluated according to the criteria listed in Appendix A [12]. In blood gas evaluation, the acid–base disorders were classified as acidosis (respiratory or metabolic) and alkalosis (respiratory or metabolic) based on the criteria, respectively. The diagnosis of acid–base disorders was based on whether all criteria items were met.

### 2.4. Statistical Analysis

Each survey item was compared based on the anatomical site involved. The results were compared for each respiratory disease at the anatomical site. Furthermore, we compared the relevance of each item between GC and GA. Analyses were performed nonparametrically using the Mann–Whitney *U* test for comparisons between two groups and the Kruskal–Wallis test and Dunn’s multiple tests for comparisons between three or more groups. All analyses were performed using statistical software (Prism, version 9.00, GraphPad Software, San Diego, CA, USA), and *p*-values < 0.05 were considered statistically significant.

## 3. Results

In total, 717 patients in GC and 1061 patients in GA with respiratory symptoms suggestive of respiratory disease were included in this study. Of these, 13 cases from GC and 379 cases from GA were excluded because the cause of the respiratory symptoms was not a respiratory disease. Therefore, our study included 704 GC patients and 682 GA patients. GC included 391 males (269 castrated) and 313 females (240 spayed), whereas GA included 307 males (203 castrated) and 375 females (289 spayed). The median age of patients with GC was 121 (range: 2–222) months and that of patients in GA was 155 (range: 1–241) months. Chihuahua was the most common breed in both GC (15.8%, n = 111) and GA (21.7%, n = 148), followed by toy poodles (13.9%, n = 98 and 12.6%, n = 86, respectively), and miniature dachshunds (12.2%, n = 86 and 12.3%, n = 84, respectively). Others included 50 purebreds and 40 crossbreeds in GC and 37 purebreds and 69 crossbreeds in GA. The details of the respiratory diseases categorized by the site of involvement in GC were as follows: trachea/bronchi (39.3%, n = 277), nasal cavity (20.7%, n = 146), pharynx (14.9%, n = 105), larynx (12.8%, n = 90), lung (10.9%, n = 77), and mediastinum or pleural cavity (1.4%, n = 9). In contrast, in the GA group, the involved sites were the lung (61.3%, n = 418), trachea/bronchi (14.7%, n = 100), laryngopharynx (14.1%, n = 96), mediastinum or pleural cavity (6.2%, n = 42), and nasal cavity (3.7%, n = 26). The details of the disease at each anatomical site in the GC and GA groups are presented in Table 1.

### 3.1. RR

Data on RR were available for 400 GC cases and 586 GA cases. A significant increase in RR was observed in GA compared to that in GC (*p* < 0.0001). In GC, the RR of lung disease was significantly higher than those of nasal, pharyngeal, laryngeal, and tracheobronchial diseases (Figure 1A). Similarly, in GA, a significant increase in RR was observed in lung disease compared to that in nasal, laryngopharyngeal, tracheobronchial, and mediastinal or pleural disease (Figure 1B). The median RR for lung disease was 43 breaths/min (range: 18–150 breaths/min) and 70 breaths/min (range: 12–216 breathes/min) in GC and GA, respectively (Table 2). Interstitial lung disease (ILD), aspiration pneumonia, lung mass, and bronchopneumonia, which were common in GC, were compared; however, no clear association was found. In contrast, in GA, when pulmonary edema (cardiogenic and non-cardiogenic), aspiration pneumonia, bronchopneumonia, and lung mass were compared, the RR was significantly greater in pulmonary edema than in aspiration pneumonia (Figure 2).

### 3.2. SpO_2_

Data on SpO_2_ were available for 410 GC patients (nasal cavity: n = 59, larynx: n = 71, larynx: n = 62, trachea/bronchi: n = 158, lung: n = 54, and mediastinum/pleural cavity: n = 6) and 529 GA patients (nasal cavity: n = 20, laryngopharynx: n = 76, trachea/bronchi: n = 82, lung: n = 323, and mediastinum/pleural cavity: n = 28). Group C showed a lower value than 95% in 9.6% (40/410), while GA showed a lower value in 51.4% (272/529). Low values of both GC and GA were most common in lung diseases, with the proportions being 31.5% (17/54) and 69.7% (225/323), respectively. The SpO_2_ in GA was significantly lower than that in GC (*p* < 0.0001), with a median of 98% (range: 80–100%) in GC and 94% (range: 60–100%) in GA (Table 2). In GC, SpO_2_ in lung disease was significantly lower than that in nasal (*p* = 0.0001) and laryngeal diseases (*p* = 0.0040), and pharyngeal (*p* = 0.0368) and tracheobronchial (*p* = 0.0394) diseases were significantly lower than nasal disease. In GA, the SpO_2_ in lung disease was significantly lower than that in nasal (*p* = 0.0023), laryngopharyngeal (*p* < 0.0001), and tracheobronchial diseases (*p* < 0.0001), and the median value for lung disease was 91% (range: 60–100%) (Table 2). Among lung diseases, SpO_2_ was significantly lower in ILD than in lung masses in GC (*p* = 0.0042), with a median SpO_2_ of 92% (Table 3). In GA, no significant association was found with any of the lung diseases, although all diseases had a median SpO_2_ of <95% (Table 3).

### 3.3. CBC

Complete blood count data were available for 541 GC cases (nasal cavity: n = 127, larynx: n = 65, larynx: n = 76, trachea/bronchi: n = 199, lung: n = 66, and mediastinum/pleural cavity: n = 8) and 547 GA cases (nasal cavity: n = 12, laryngopharynx: n = 56, trachea/bronchi: n = 66, lung: n = 379, and mediastinum/pleural cavity: n = 34). In addition, leukocyte fractionation data from 185 GC and 282 GA cases were available. In GC as a whole, the percentages of RBC, HCT, HGB, WBC, and platelet that were below the reference values were 11.5% (62/541), 8.1% (44/541), 6.7% (36/541), 5.5% (30/541), and 8.7% (47/541), and the percentages that were above the reference values were 4.6% (25/541), 8.7% (47/541), 13.7% (74/541), 20.5% (111/541), and 31.8% (172/541), respectively. In contrast, for GA as a whole, 17.7% (97/547), 18.8% (103/547), 14.4% (79/547), 2.9% (16/547), and 6.8% (37/547) were below the reference values, and 9.3% (51/547), 9.5% (52/547), 17.0% (93/547), 41.5% (227/547), and 38.8% (212/547) were above it, respectively. In GC, decreases in RBC, HCT, and HGB tended to occur more often in lung disease (13/66, 19.7%, 11/66, 16.7%, and 10/66, 15.2%, respectively), but no clear differences were observed in increases in these items among disease sites. Additionally, increases in WBC tended to occur more frequently in patients with lung (29/66, 43.9%) and mediastinal or pleural diseases (4/8, 50.0%). In contrast, in GA, decreases in RBC, HCT, and HGB tended to occur more often in mediastinal or pleural diseases (10/34, 29.4%, 11/34, 32.4%, and 10/34, 29.4%, respectively), and increases in WBC tended to occur more often in lung (177/379, 46.7%), mediastinal or pleural (14/34, 41.2%), and nasal diseases (6/12, 50.0%). Among the CBC items, HCT (*p* = 0.0002) and HGB (*p* = 0.0063) levels were significantly higher in GC, and WBC count was significantly higher in GA (*p* < 0.0001). There were no significant differences in RBC and platelet counts between the groups. In addition, lymphocyte (*p* = 0.0094) and eosinophil (*p* < 0.0001) counts were significantly higher in GC, whereas segmented neutrophil (*p* < 0.0001) and monocyte (*p* = 0.0010) counts were significantly higher in GA. In GC, HGB levels were significantly lower in patients with lung disease than in those with nasal (*p* = 0.0102), pharyngeal (*p* = 0.0181), laryngeal (*p* = 0.0158), and tracheobronchial (*p* = 0.0127) diseases. The WBC count was significantly higher in lung disease than in pharyngeal (*p* < 0.0001), laryngeal (*p* < 0.0001), and tracheobronchial (*p* = 0.0008) diseases, and that in mediastinal or pleural disease was significantly higher than that in pharyngeal (*p* = 0.0208) and laryngeal (*p* = 0.0071) diseases. The median WBC increased to 17,150/μL (range: 10,700–34,600/μL) with mediastinal or pleural disease but was within the reference range with lung disease at 15,050/μL (range: 3800–101,100/μL) (Table 2). The leukocyte cell fraction showed significantly more segmented neutrophils in lung disease than in laryngeal disease (*p* = 0.0154) and a significant increase in monocytes in lung disease compared to pharyngeal (*p* = 0.0263), laryngeal (*p* = 0.0003), and tracheobronchial (*p* = 0.0005) diseases. In addition, the median counts of segmented neutrophils and monocytes in lung disease increased to 15,096/μL (range: 5976–92,507/μL) and 1573/μL (range: 350–7077/μL), respectively (Table 2). In GA, there was a significant increase in the WBC count in lung disease when compared with the counts in laryngopharyngeal (*p* = 0.0027) and tracheobronchial (*p* < 0.0001) diseases, and in mediastinal or pleural disease as compared to tracheobronchial disease (*p* = 0.0318). However, the median WBC count was 16,400/μL (range: 1200–60,000/μL) in lung disease and 16,350/μL (range: 7400–73,000/μL) in mediastinal or pleural disease, which is within the reference range (Table 2). Considering leukocyte fractionation, the segmented neutrophils were considerably more in lung disease than in tracheobronchial disease (*p* = 0.0099), with a median value of 16,544/μL (range: 516–54,126/μL) (Table 2). Among lung diseases, the WBC count was significantly higher in patients with bronchopneumonia than in those with pulmonary edema (*p* = 0.0134).

### 3.4. Blood Gas Analysis

Blood gas analysis data were available for 378 GC (nasal cavity: n = 92, larynx: n = 69, larynx: n = 60, trachea/bronchi: n = 100, lung: n = 51, and mediastinum/pleural cavity: n = 6) and 504 GA cases (nasal cavity: n = 8, laryngopharynx: n = 42, trachea/bronchi: n = 39, lung: n = 382, and mediastinum/pleural cavity: n = 33), with venous blood findings in all GC cases, arterial blood in 139 GA cases, and venous blood in 365 GA cases. The percentages of pH, PCO_2_, HCO_3_, and BE that were below the reference values in the entire GC were 11.4% (43/378), 63.5% (240/378), 38.6% (146/378), and 34.7% (131/378), respectively, and those that were above values were 59.8% (226/378), 18.0% (68/378), 26.6% (99/378), and 27.8% (105/378), respectively. In contrast, in GA, the percentages that were below were 47.7% (239/504), 43.3% (218/504), 47.6% (240/504), and 53.2% (268/504), and the percentages that were above were 25.6% (129/504), 39.1% (197/504), 21.6% (109/504), and 18.3% (92/504), respectively. Comparison of venous blood between the groups revealed significant differences in pH, PCO_2_, HCO_3_ concentration, and BE between GC and GA (*p* < 0.0001). Group C had a significantly higher pH and lower PCO_2_ than GA, whereas GA had a significantly lower HCO_3_ concentration and BE than GC. By site of disease in GC, the pH in pharyngeal disease was significantly lower than in nasal disease (*p* = 0.0022), and its PCO_2_ was significantly higher than that in nasal (*p* = 0.0003) and tracheobronchial (*p* = 0.0372) diseases. Brachycephalic obstructive airway syndrome (BOAS) showed a significantly lower pH and higher PCO_2_ than those of other diseases, with median values of 7.395 (range: 7.17–7.50) and 42.5 mmHg (range: 25–72 mmHg), respectively. In GA, the pH, PCO_2_, HCO_3_ concentration, and BE were not significantly different between the anatomical sites for both arterial and venous blood. However, laryngopharyngeal and lung disease tended to have a lower pH (7.33 and 7.34, respectively) and higher PCO_2_ (both 44 mmHg) than those of other diseases (Table 2). Pulmonary edema was associated with a significantly lower venous blood pH than aspiration pneumonia (*p* = 0.0008), and PCO_2_ in pulmonary edema and bronchopneumonia was significantly higher than that in aspiration pneumonia (*p* = 0.0003 and 0.0174, respectively). In pulmonary edema and bronchopneumonia, the median pH was 7.32 (range: 6.80–7.59) and 7.29 (range: 7.03–7.48), respectively, and the median PCO_2_ was 47 mmHg (range: 21–139 mmHg) and 46 mmHg (range: 29–81 mmHg), respectively (Table 3). No characteristic trends were observed in arterial blood at any disease site.

Acidosis or alkalosis was detected in 71.2% (269/378) of patients in GC and 71.2% (359/504) of patients in GA, as assessed by blood gas analysis. In acid–base disorders in GC, respiratory alkalosis (64.3%, n = 173) was the most common, followed by metabolic alkalosis (19.7%, n = 53), respiratory acidosis (8.6%, n = 23), and metabolic acidosis (6.7%, n = 18), whereas in GA, respiratory acidosis (29.8%, n = 107) was the most common, followed by respiratory alkalosis (27.8%, n = 100), metabolic acidosis (23.4%, n = 84), and metabolic alkalosis (7.0%, n = 25) (Table 4). Respiratory alkalosis was the most common anatomical site in GC (Table 4). Respiratory acidosis in GC was found in 6.0% (16/269) of patients with laryngopharyngeal disease and 2.2% (6/269) of those with tracheobronchial disease, but not in those with lung, mediastinal or pleural disease. However, GA was found in 24.8% (89/359) of the lung disease cases and in 2.0% (7/359) of the mediastinal or pleural disease cases. Metabolic alkalosis and acidosis showed no significant differences in incidence between the anatomical sites for either GC or GA.

### 3.5. CRP

The CRP levels were available for 541 GC and 530 GA cases. GA had a significantly higher CRP level than that of GC (*p* < 0.0001), with a median of 0.47 μg/mL (range: 0–16.56 μg/mL) in GC and 2.10 μg/mL (range: 0–21.0 μg/mL) in GA (Table 4). In GC, the CRP level in lung disease was significantly higher than that in pharyngeal, laryngeal, and tracheobronchial diseases (Figure 3A), but the median value was 0.82 μg/mL (range: 0.02–16.56 μg/mL) (Table 2). Nasal disease also resulted in a significant increase in CRP levels compared to pharyngeal and laryngeal diseases (Figure 3A). In contrast, in GA, the CRP level in lung disease was significantly higher than that in laryngopharyngeal and tracheobronchial diseases (Figure 3B), with a median value of 2.6 μg/mL (range: 0–21.0 μg/mL) (Table 4). A comparison of lung diseases in GC showed significantly higher CRP levels in bronchopneumonia than in ILD (*p* = 0.0268), but not in other diseases. The median CRP level increased to 4.85 μg/mL (range: 1.19–16.56 μg/mL) in bronchopneumonia (Table 3). In addition, there was no significant difference in CRP levels in nasal diseases in GC. In GA, among lung diseases, the CRP levels in bronchopneumonia and aspiration pneumonia were significantly greater than those in pulmonary edema (cardiogenic and non-cardiogenic), with median values of 4.0 μg/mL (range: 0.3–14.3 μg/mL) and 2.6 μg/mL (range: 0.3–13.1 μg/mL), respectively (Table 3).

## 4. Discussion

This study investigated the relationship between RR, SpO_2_, and various blood tests commonly used in veterinary clinical practice, and respiratory diseases. Although the present results showed statistically significant associations between some items and respiratory diseases, some were not considered clinically useful based on the measured values. An increase in RR was significantly associated with lung disease in both GC and GA. This result is consistent with previous reports [1]. Among the lung diseases, no significant difference was observed in GC, whereas in GA lung diseases, the median RR showed an abnormal value. In addition, the RR in GA was significantly greater for both overall and lung diseases than that in the GC. Therefore, the RR can provide information on a patient’s acute-phase condition and the presence of lung disease. However, because the RR of GC was also significantly increased in lung disease, it is necessary to consider the increase in RR due to chronic hypoxemia. In addition, an increased RR can be caused by conditions other than respiratory diseases. In clinical practice, panting is often encountered owing to excitement, heat, and stress. Panting is a compensatory behavior that involves opening the mouth and rapid, shallow breathing without writhing and should therefore be carefully differentiated [20]. Other factors include pain and cardiovascular disease; therefore, in cases of increased RR, a comprehensive evaluation in conjunction with various tests, such as blood tests and radiography, is necessary.

Evaluating SpO_2_ enables rapid assessment of arterial oxygenation without blood sampling. One of the most important reasons for assessing arterial oxygenation in veterinary practice is to detect hypoxemia. Hypoxemia has been reported to occur mainly in lung and upper airway-obstructive respiratory diseases [21,22,23]. In this study, a significant decrease in SpO_2_ was observed in both GC and GA in lung diseases compared with other diseases. Specifically, in GA, the median SpO_2_ for lung diseases was 91%, which was below the reference range, and the median SpO_2_ for all lung diseases in GA was <95%. In contrast, a significant decrease in lung disease was observed in GC, although the median value was 97%, which was within the reference range. Among lung diseases, a decrease in SpO_2_ was observed, with a median value of 91% in the ILD group. Chronic hypoxemia has been reported to occur in various lung diseases such as ILD (e.g., canine idiopathic pulmonary fibrosis) and lung mineralization [24,25]. However, the number of ILD cases in this study was small, and we could not determine whether SpO_2_ was clinically relevant to ILD. In addition, SpO_2_ is influenced by various factors and is considered less accurate than PaO_2_. Therefore, based on the results of this study, SpO_2_ may serve as an auxiliary testing tool for assessing and monitoring respiratory conditions in patients with acute and chronic lung diseases that cause chronic hypoxia. However, to determine whether SpO_2_ is clinically useful, it is necessary to investigate the correlation between SpO_2_ and PaO_2_, and these diseases.

In the CBC, especially the WBC count, both GC and GA showed significant increases in lung and mediastinal or pleural diseases. An increase in WBC count is reported to occur due to inflammation, stress, hypersensitivity, or neoplasia [13,26]. In this study, the neutrophil (segmented) and monocyte counts in GC were higher than those in GA. In GA, the neutrophil (segmented) count was increased in lung disease. Neutrophilia occurs in inflammation, neoplasia, glucocorticoid-associated diseases, and excitement, especially in inflammation, and its extent varies from mild to severe, depending on the severity [27]. Therefore, the increase in segmented neutrophils in this study suggests that inflammation of the lung is a more severe condition than inflammation of the other sites within the respiratory system. In addition to inflammation, the use of corticosteroids may have influenced the increase in neutrophils and monocytes observed in GC. At facilities targeting GC, all patients were evaluated by a referral hospital for a second opinion. Therefore, we included cases in which glucocorticoid therapy had already been initiated, which may have contributed to the increase in neutrophils and monocytes. However, the median values of the WBC count for lung and mediastinum or pleural disease in GC and lung disease in GA were within the reference range, and the clinical usefulness could not be determined. The non-leukocyte parameters of HGB were significantly decreased in patients with lung disease in GC. In this study, the median HGB level in patients with lung disease was within the reference range, suggesting no association between lung disease and anemia. In humans, chronic inflammation due to chronic lung disease is thought to be the underlying cause of anemia and has already been well-characterized in patients with chronic obstructive pulmonary disease [28]. In dogs, prolonged tissue hypoxia caused by chronic lung disease is generally considered an important mechanism for the development of secondary polycythemia [29]. Although several studies have investigated the prevalence of polycythemia in dogs with chronic hypoxemia, its occurrence is rare [30,31]. In this study, no apparent increase in erythrocytes was observed in the lung disease group, suggesting that a pathology different from that in humans was involved.

All cases of GC were evaluated using venous blood, so a comparison between GC and GA using arterial blood could not be performed. Additionally, because there was some disease bias in this study, it was not possible to examine the relationship between PaO_2_ and SpO_2_. Blood gas analysis using venous blood revealed a difference in the occurrence of acid–base disorders between GC and GA. Respiratory alkalosis was the most common cause of GC, accounting for 64.3% of the cases. Respiratory alkalosis refers to alveolar hyperventilation that leads to a decrease in carbon dioxide [4]. The causes of alveolar hyperventilation include hypoxemia, compensatory reactions to metabolic acidosis, abnormal control of respiration in the central nervous system, and panting [4,32]. In this study, there was no difference in the incidence of respiratory alkalosis between the GC sites, and the occurrence of metabolic acidosis was rare. Therefore, in addition to the presence of diseases such as hypoxia and respiratory center abnormalities, hyperventilation in GC may also be related to an increase in RR due to various factors such as excitement and heat. The incidence of respiratory alkalosis in GA was 27.8%, lower than that in GC, and respiratory acidosis was the most common abnormality. The causes of respiratory acidosis include various factors that decrease ventilation, such as anesthesia, airway-obstructive disease, and lung disease [4,33]. No significant differences in pH or PCO_2_ were found between the different GA sites; however, respiratory acidosis tended to occur more frequently in lung, mediastinal or pleural, and laryngopharyngeal diseases. These results support the previously reported causes of respiratory acidosis [4,33]. However, respiratory acidosis due to lung disease was most common in GA but not in GC. Therefore, a compensatory mechanism might be involved in this process. Metabolic compensatory mechanisms that increase the HCO_3_ concentration are involved in respiratory acidosis. However, while the respiratory compensatory mechanisms occur immediately, the metabolic compensatory mechanism occurs over a period of several days or longer [9,34]. Therefore, it is thought that the compensatory mechanism does not function in cases of respiratory acidosis in the acute phase and that there are more cases in the acute phase than in the chronic phase. Laryngopharyngeal disease is a non-pulmonary cause of respiratory acidosis in both GA and GC. In the acute phase, it is thought to develop due to transient upper airway obstruction, similar to that observed in lung disease. However, the factors that cause respiratory acidosis, even in the chronic phase, are not limited to the acute exacerbation of chronic hypoxia but may also be affected by the breed. In this study, BOAS showed significantly lower pH and higher PCO_2_ values in GC than in other diseases. A previous study reported that normal brachycephalic breeds have significantly higher PCO_2_ in the arterial blood than mesocephalic or dolichocephalic breeds [21,23]. Therefore, care must be taken when evaluating brachycephalic breeds, regardless of the presence or absence of disease, and the reference values must be reconsidered.

In the present study, the CRP level was significantly elevated in GA compared to GC, and across disease localization, both groups showed significantly increased CRP levels in lung disease. Previous studies have investigated the relationship between various diseases and CRP levels [14]. However, because of its low specificity, CRP level should not be regarded as a marker for etiology diagnosis. Its usefulness in evaluating the presence of inflammation and monitoring treatment response has been reported [15]. In lung diseases, it has been found to be associated with bacterial pneumonia, bronchopneumonia, and lung masses [14,16] and has been reported to be useful for monitoring bacterial pneumonia and aspiration pneumonia [17,35]. Elevated CRP levels have been observed in various respiratory diseases, and this result is useful for assisting diagnosis and monitoring. However, because CRP levels are also elevated in various non-respiratory diseases, it is necessary to make a comprehensive diagnosis in conjunction with other tests.

This study has some limitations. First, there were variations in the technique, time, and site of blood sampling. Because this was a retrospective study, only numerical values were recorded based on the medical database. Therefore, the person who collected the blood and the site from which the blood was collected were not recorded. In blood gas analysis, it has been reported that the numerical value varies depending on the sampling site [11]. In this study, blood was primarily collected from the jugular vein; however, in case of animal tolerance or emergencies, blood was collected from the saphenous or cephalic vein. To investigate the relationship between accurate blood test values and diseases, it is necessary to standardize blood sampling sites. Second, because this was a retrospective study, the diagnoses were made by multiple people, and cases without a definitive diagnosis were included. Especially in the acute phase, there were many deaths due to worsening respiratory conditions or those in which the owners did not want further examination. Another limitation is the influence of various factors on SpO_2_. In this study, SpO_2_ measurements were performed on the ears, lips, and tail, depending on various factors such as tolerance, respiratory conditions, and coat color. It is possible that the measurement conditions and sites were not standardized, which may have affected the measured values. Finally, duplicate patients with comorbidities were included. Therefore, factors other than the underlying disease may have affected the results of each test.

## 5. Conclusions

In conclusion, this study found an association among various test values, respiratory sites, and disease. Although a statistically significant relationship was observed for some test items, whether they are clinically useful could not be determined. Therefore, these tests may be used as adjuncts to other tests such as imaging for the diagnosis and monitoring of treatment responses. Furthermore, based on the results of this study, we believe that further investigation into the relationship with clinical symptoms and other tests such as body temperature of respiratory diseases could provide even more useful information. In this study, the correlation between the test values was not investigated. In particular, it is necessary to investigate the relationship between PaO_2_ and SpO_2_, which could not be investigated due to disease bias or the inability to measure arterial blood gas in GC. In addition, it is necessary to evaluate the correlation of fluctuations among various tests and further investigate the trends and usefulness of these tests for various respiratory diseases.

## Figures and Tables

**Figure 1 vetsci-11-00027-f001:**
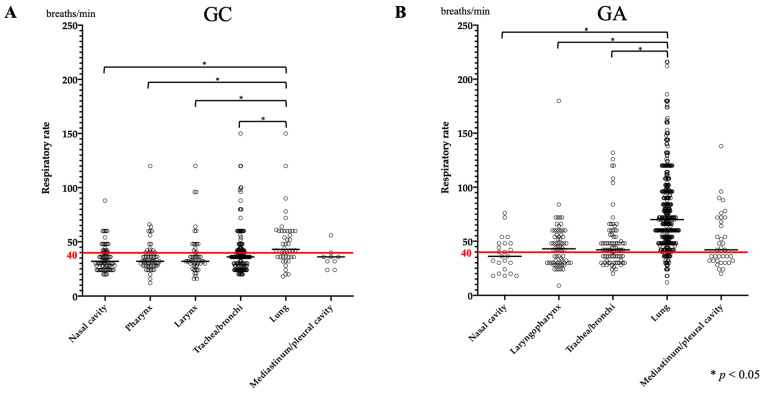
Comparison of respiratory rate between anatomical sites in GC and GA. (**A**) The respiration rate in GC showed a significant increase in lung disease with a median of 43 breaths/min. (**B**) In GA, respiration rate was also significantly increased in lung disease with a median of 70 breaths/min. However, the median value was also above the reference value at other sites except the nasal cavity. The red line shows the reference value.

**Figure 2 vetsci-11-00027-f002:**
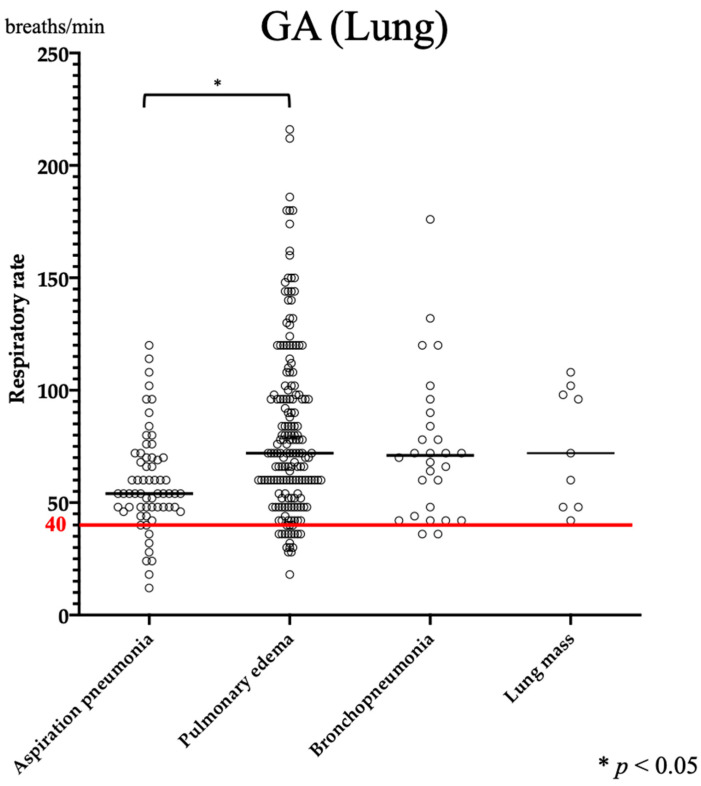
Comparison of respiratory rate among lung diseases in GA. There was a significant increase in pulmonary edema compared to aspiration pneumonia, with a median value > 40 breaths/min for all diseases. The red line shows the reference value.

**Figure 3 vetsci-11-00027-f003:**
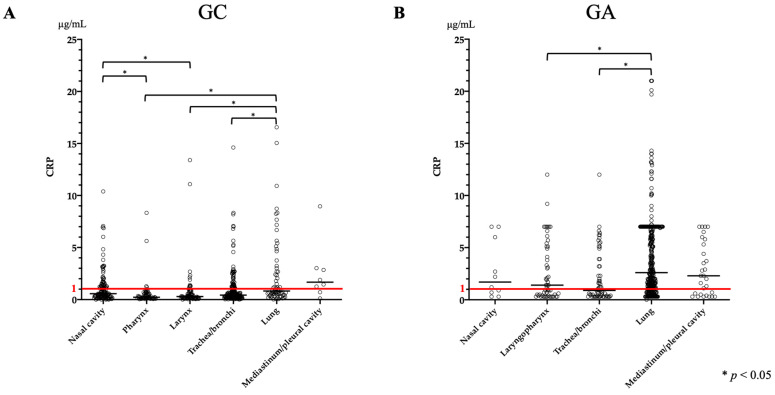
Comparison of CRP between anatomical sites in GC and GA. (**A**) Elevations were noted in lung and nasal disease. Mediastinum or pleural disease also had a median increase but was not associated. (**B**) There was a significant increase in lung disease, but the median value was increased in all sites except the trachea/bronchi. The red line shows the reference value.

**Table 1 vetsci-11-00027-t001:** Classification of respiratory diseases based on the anatomical site in GC and GA.

Anatomical Sites and Final Diagnosis	Number of Dogs in GC (%)	Number of Dogs in GA (%)
Total	704	682
Nasal cavity	146 (20.7%)	26 (3.7%)
Non-infectious rhinitis	56 (8.0%)	0
Neoplasia	47 (6.7%)	1 (0.1%)
Foreign body	21 (2.9%)	0
Rhinitis secondary to dental disease	16 (2.3%)	0
Fungal rhinitis	5 (0.7%)	0
Rhinitis (cause unknown) *	0	22 (3.2%)
Other diseases	1 (0.1%)	3 (0.4%)
Laryngopharynx ^†^	195 (27.7%)	96 (14.1%)
BOAS	98 (13.9%)	2 (0.3%)
Laryngopharyngitis	16 (2.3%)	45 (6.6%)
Upper airway obstruction (cause unknown)	0	35 (5.1%)
Laryngeal paralysis	28 (4.0%)	5 (0.8%)
Laryngeal collapse	13 (1.8%)	0
Other diseases	40 (5.7%)	9 (1.3%)
Trachea/bronchi	277 (39.3%)	100 (14.7%)
Tracheobronchial collapse	135 (19.2%)	34 (5.0%)
Chronic bronchitis	120 (17.0%)	0
Bronchitis	0	53 (7.8%)
Bronchiectasis	10 (1.4%)	1 (0.1%)
Infectious bronchitis	7 (1.0%)	12 (1.8%)
Other diseases	5 (0.7%)	0
Lung	77 (10.9%)	418 (61.3%)
Cardiogenic pulmonary edema	0	172 (25.2%)
Pneumonia (cause unknown) *	0	83 (12.2%)
Aspiration pneumonia	19 (2.7%)	66 (9.7%)
Bronchopneumonia	6 (0.9%)	29 (4.2%)
Non-cardiogenic pulmonary edema	1 (0.1%)	28 (4.1%)
Lung masses	25 (3.6%)	10 (1.5%)
Interstitial lung disease	18 (2.5%)	2 (0.3%)
Other diseases	8 (1.1%)	28 (4.1%)
Mediastinum/pleural Cavity	9 (1.4%)	42 (6.2%)
Pericardial effusion	0	11 (1.6%)
Pleural effusion	3 (0.5%)	6 (0.8%)
Anterior mediastinum tumor	4 (0.6%)	2 (0.3%)
Pulmonary hypertension	0	5 (0.7%)
Other diseases	2 (0.3%)	19 (2.8%)

BOAS, brachycephalic obstructive airway syndrome; GA, acute disease group; GC, chronic disease group. * Dogs diagnosed with rhinitis and pneumonia included dogs with an unknown cause of disease for which a provisional or definitive diagnosis could not be made owing to factors such as signalment, history, various tests, response to therapy, or death before diagnosis. ^†^ Pharyngeal and laryngeal diseases in the acute phase were classified as laryngopharyngeal diseases because there were many cases in which the cause could not be identified by computed tomography or laryngeal endoscopy, and many patients died before diagnosis. For consistency, laryngopharyngeal disease was used as a defined subclassification in GC.

**Table 2 vetsci-11-00027-t002:** Details of test result values for each anatomical site in Group C and A.

	Median Value (Range)						
GC	Total	Nasal Cavity	Pharynx	Larynx	Trachea/Bronchi	Lung	Mediastinum/Pleural Cavity
Respiratory rate (/min)	36 (12–150)	32 (20–88)	32 (12–120)	32 (16–120)	36 (20–150)	**43** (18–150)	36 (24–56)
SpO_2_ (%)	98 (80–100)	99 (93–100)	98 (90–100)	98 (83–100)	98 (90–100)	97 (80–100)	98 (95–100)
CBC							
Red blood cell (/μL)	6.79 × 10^6^ (2.70–10.09)	6.97 × 10^6^ (2.99–9.67)	6.64 × 10^6^ (3.62–8.67)	6.72 × 10^6^ (2.70–8.72)	6.85 × 10^6^ (4.60–10.09)	6.34 × 10^6^ (4.54–9.83)	6.62 × 10^6^ (3.34–7.98)
Hematocrit (%)	46.9 (21.9–64.0)	47.5 (23.1–62.8)	47.7 (26–61.7)	47.1 (33.5–59.5)	46.9 (32.0–60.9)	45.2 (29.7–64.0)	44.9 (21.9–54.5)
Hemoglobin (g/dL)	15.8 (6.8–70.2)	16.2 (7.3–70.2)	15.9 (8.6–21.3)	16.2 (10.9–21.0)	15.8 (9.8–23.4)	15.1 (9.5–21.6)	15.3 (6.8–18.1)
White blood cell (/μL)	11,650 (2400–101,100)	12,200 (4800–52,000)	10,800 (4400–25,400)	9600 (4400–34,600)	11,300 (2400–47,500)	15,050 (3800–101,100)	17,150 (10,700–34,600)
Segmented neutrophil (/μL)	**11,822** (1154–92,507)	10,875 (1154–46,810)	9792 (3476–23,368)	8298 (3276–32,752)	**11,700** (2204–43,225)	**15,096** (5976–92,507)	**12,060** (11,822–32,752)
Platelet (/μL)	423 × 10^3^ (9–1430)	415 × 10^3^ (41.4–1195)	423 × 10^3^ (64.2–810)	442 × 10^3^ (88–902)	408 × 10^3^ (9–1430)	453 × 10^3^ (34–1280)	452 × 10^3^ (290–720)
Blood gas analysis							
pH	**7.42** (7.17–7.55)	**7.43** (7.35–7.55)	7.40 (7.17–7.51)	**7.42** (7.17–7.54)	**7.42** (7.26–7.54)	**7.42** (7.33–7.54)	**7.46** (7.36–7.50)
PCO_2_ (mmHg)	**38** (18–72)	**35.5** (19–52)	41 (19–72)	**38** (22–72)	**37** (18–68)	**39** (27–48)	**35** (30–47)
HCO_3_ (mmol/L)	24.2 (14.4–40.0)	23.5 (15.9–35.9)	25.1 (15.2–36.4)	23.6 (18.4–40.0)	24.2 (14.4–38.7)	25.2 (16.0–33.5)	23.9 (21.3–26.8)
BE (mmol/L)	−0.4 (−11.4–14.1)	−1.1 (−8–13.2)	0.3 (−7.8–11.8)	−0.4 (−7.2–13.2)	−0.6 (−11.4–14.1)	0.8 (−9.1–10.6)	0.3 (−2.5–3.3)
CRP (μg/mL)	0.47 (0–16.56)	0.57 (0–10.39)	0.22 (0.04–8.33)	0.29 (0–13.4)	0.43 (0–14.61)	0.82 (0.02–16.56)	**1.67** (0.11–8.96)
**GA**	**Total**	**Nasal Cavity**	**Laryngopharynx**	**Trachea/Bronchi**	**Lung**	**Mediastinum/Pleural Cavity**	
Respiratory rate (/min)	**60** (9–216)	36 (9–216)	**43** (9–72)	**42** (20–132)	**70** (12–216)	**42** (20–138)	
SpO_2_ (%)	**93** (60–100)	96 (60–100)	97 (75–100)	96 (71–100)	**91** (60–100)	95 (80–100)	
CBC							
Red blood cell (/μL)	6.92 × 10^6^ (2.62–11.65)	6.98 × 10^6^ (3.09–10.68)	6.99 × 10^6^ (3.38–9.46)	7.03 × 10^6^ (3.90–10.35)	6.92 × 10^6^ (2.62–11.65)	6.38 × 10^6^ (3.01–10.57)	
Hematocrit (%)	45.6 (16.8–73.7)	45.4 (21.7–73.5)	47.1 (26.4–65.0)	46.3 (24.3–64.0)	45.6 (16.8–73.7)	42.1 (22.6–67.3)	
Hemoglobin (g/dL)	15.6 (5.7–25.7)	15.6 (7.3–24.4)	16.2 (8.8–22.8)	15.7 (8.1–21.4)	15.6 (5.7–25.7)	14.8 (7.5–22.8)	
White blood cell (/μL)	15,300 (840–73,000)	**18,150** (5900–42,400)	12,100 (840–56,500)	12,650 (4100–44,500)	16,400 (1200–60,000)	16,350 (7400–73,000)	
Segmented neutrophil (/μL)	**15,747** (516–68,620)	**19,888** (4968–24,055)	**14,233** (2408–50,850)	11,340 (3034–32,930)	**16,544** (516–54,126)	**13,689** (6992–68,620)	
Platelet (/μL)	436 × 10^3^ (0–1330)	514 × 10^3^ (216–716)	383 × 10^3^ (115–947)	404 × 10^3^ (115–857)	455 × 10^3^ (0–1330)	342 × 10^3^ (51–887)	
Blood gas analysis (Vein)							
pH	**7.35** (6.80–7.59)	**7.41** (7.29–7.49)	**7.33** (7.13–7.47)	7.38 (7.13–7.50)	**7.34** (6.80–7.59)	**7.35** (6.95–7.56)	
PCO_2_ (mmHg)	44 (20–139)	**30** (28–31)	44 (26–72)	43 (27–90)	44 (21–139)	41 (20–59)	
HCO_3_ (mmol/L)	**22.6** (8.3–42.5)	**19.3** (13.5–22.1)	23.2 (13.7–29.5)	24.7 (16.7–31.2)	23.7 (12.1–42.5)	23.4 (8.3–31.3)	
BE (mmol/L)	**−2.4** (−21.0–16.9)	**−5.5** (−13.1–1.2)	**−2.0** (−12.7–4.2)	−0.4 (−9.7–7.0)	**−2.1** (−21.0–16.9)	−1.6 (−20.9–7.6)	
CRP (μg/mL)	**2.1** (0–21.0)	**2.7** (0.3–7.0)	**1.4** (0.3–12.0)	0.9 (0.3–12.0)	**2.6** (0–21.0)	**2.3** (0.3–7.0)	

Items that deviate from the reference value are in bold.

**Table 3 vetsci-11-00027-t003:** Details of respiratory rate, pulse oximetry, white blood cell, blood gas analysis, and C-reactive protein measurements in lung diseases of Groups C and A.

Inspection Item	Median Value (Range)			
GC	Interstitial Lung Disease	Lung Mass	Aspiration Pneumonia	Bronchopneumonia
Respiratory rate (/min)	**50** (20–150)	**44** (20–72)	40 (32–120)	**60** (56–61)
SpO_2_ (%)	**91** (83–100)	98 (90–100)	96 (80–100)	97 (90–98)
White blood cell (/μL)	15,600 (8400–50,800)	14,500 (3800–101,100)	15,300 (6300–29,700)	**22,400** (14,600–36,000)
Segmented neutrophil (/μL)	**16,248** (7189–52,490)	**12,994** (5976–92,507)	**17,436** (10,777–27,027)	**19,712** (12,647–32,400)
Blood gas analysis				
pH	**7.43** (7.38–7.49)	**7.41** (7.34–7.50)	**7.42** (7.33–7.52)	7.39 (7.35–7.54)
PCO_2_ (mmHg)	**39** (29–46)	**35** (27–41)	42 (36–48)	**35.5** (31–40)
HCO_3_ (mmol/L)	26.5 (18.0–32.0)	**22.4** (16.0–27.8)	26.3 (19.0–33.5)	**22.5** (20.1–26.5)
BE (mmol/L)	**2.3** (−6.8–8.0)	−1.8 (−9.1–3.6)	1.5 (−6.9–10.6)	**−2.7** (−5.0–4.0)
CRP (μg/mL)	0.64 (0.02–5.65)	**1.14** (0.05–10.92)	0.71 (0.20–15.05)	4.85 (1.19–16.56)
**GA**	**Bronchopneumonia**		**Aspiration Pneumonia**	**Pulmonary Edema**
Respiratory rate (/min)	**60** (9–216)		36 (9–216)	**43** (9–72)
SpO_2_ (%)	**90** (67–99)		**91** (67–100)	**92** (64–100)
White blood cell (/μL)	**17,500** (11,100–39,400)		16,800 (2000–60,000)	14,600 (1200–60,000)
Segmented neutrophil (/μL)	**15,708** (8769–34,145)		**16,544** (1120–53,400)	**15,327** (516–50,850)
Blood gas analysis (Vein)				
pH	**7.29** (7.03–7.48)		**7.39** (6.86–7.49)	**7.32** (6.80–7.59)
PCO_2_ (mmHg)	**46** (29–81)		**38** (26–83)	**47** (21–139)
HCO_3_ (mmol/L)	24.05 (18.7–30.8)		**22.1** (12.1–32.0)	24.0 (13.0–39.2)
BE (mmol/L)	−1.5 (−9.5–4.6)		**−2.4** (−18.7–8.0)	**−2.15** (−21.0–13.6)
CRP (μg/mL)	**4.0** (0.3–14.3)		**2.6** (0.3–13.1)	**1.6** (0–21.0)

Items that deviate from the reference value are in bold.

**Table 4 vetsci-11-00027-t004:** Details of acid–base disorders in 269 dogs in GC and 306 dogs in GA with acidemia.

Acid-Base Disorders	Number of Dogs in GC (%)	Number of Dogs in GA (%)
Total	269	359
Acidosis	43 (16.0%)	234 (65.2%)
Metabolic	18 (6.7%)	84 (23.4%)
Nasal cavity	0	4 (1.1%)
Laryngopharynx	9 (3.3%)	7 (2.0%)
Trachea/bronchi	5 (1.9%)	7 (2.0%)
Lung	4 (1.5%)	59 (16.3%)
Mediastinum/pleural cavity	0	7 (2.0%)
Respiratory	23 (8.6%)	107 (29.8%)
Nasal cavity	1 (0.4%)	0
Laryngopharynx	16 (6.0%)	8 (2.2%)
Trachea/bronchi	6 (2.2%)	3 (0.8%)
Lung	0	89 (24.8%)
Mediastinum/pleural cavity	0	7 (2.0%)
Metabolic and respiratory	2 (0.7%)	43 (12.0%)
Nasal cavity	0	0
Laryngopharynx	2 (0.7%)	2 (0.6%)
Trachea/bronchi	0	0
Lung	0	39 (10.8%)
Mediastinum/pleural cavity	0	2 (0.6%)
Alkalosis	226 (84.0%)	125 (34.8%)
Metabolic	53 (19.7%)	25 (7.0%)
Nasal cavity	11 (4.1%)	0
Laryngopharynx	17 (6.3%)	1 (0.3%)
Trachea/bronchi	13 (4.8%)	2 (0.6%)
Lung	12 (4.5%)	20 (5.5%)
Mediastinum/pleural cavity	0	2 (0.6%)
Respiratory	173 (64.3%)	100 (27.8%)
Nasal cavity	51 (19.0%)	4 (1.1%)
Laryngopharynx	49 (18.2%)	10 (2.8%)
Trachea/bronchi	49 (18.2%)	13 (3.6%)
Lung	20 (7.4%)	65 (18.1%)
Mediastinum/pleural cavity	4 (1.5%)	8 (2.2%)

## Data Availability

The data are contained within this article.

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
