# Peer review of "Relationship between Respiratory Rate, Oxygen Saturation, and Blood Test Results in Dogs with Chronic or Acute Respiratory Disease: A Retrospective Study"

_vetsci, 2024, doi:10.3390/vetsci11010027_

Round 1
Reviewer 1 Report
Comments and Suggestions for Authors
The focus of this study was to explain how many respiratory diseases can be differentiated based on respiratory rate and oxygen saturation, and by adding blood tests to these two parameters if possible, in emergency settings. I think it contains data that is extremely clinically valuable. However, I think this paper requires some modifications as follows.
In this study, the two facilities are simply called GC and GA groups, but each group contains very heterogeneous cases, making the grouping very rough. Normally, if blood gas data is available, acute and chronic conditions are classified based on pH.
However, the GC group includes cases with acidemia and alkalemia. On the other hand, the GA group may include cases with normal pH.
In addition, it also provides the definition and distribution of respiratory symptoms, which is the key-point of this paper.
To investigate the association between RR, SpO2, and various blood test findings in respiratory diseases, authors should first show the relationship between the combined these data of all cases in the GA and GC groups and each respiratory symptoms in diseases.
And, to compare these findings in chronic and acute diseases, authors should discuss the results by comparing cases with normal pH from the GC (as the actual GC) group and cases with abnormal pH from the GA (as the actual GA) group.
Comments on the Quality of English LanguagePlural and singular expressions should be unified in Table 1.
The column headings for the classification of Anatomical sites and final diagnosis are difficult to understand in Table 3.
Author Response
We are sending a manuscript above mentioned after revisions according to the reviewers’ comments. Revised sentences and words are highlighted in red in the manuscript. We appreciate helpful suggestions offered by reviewers. We hope the revised manuscript is acceptable for publication in MDPI Veterinary Science.The focus of this study was to explain how many respiratory diseases can be differentiated based on respiratory rate and oxygen saturation, and by adding blood tests to these two parameters if possible, in emergency settings. I think it contains data that is extremely clinically valuable. However, I think this paper requires some modifications as follows.
We appreciate your comment.
In this study, the two facilities are simply called GC and GA groups, but each group contains very heterogeneous cases, making the grouping very rough. Normally, if blood gas data is available, acute and chronic conditions are classified based on pH.
However, the GC group includes cases with acidemia and alkalemia. On the other hand, the GA group may include cases with normal pH.
In addition, it also provides the definition and distribution of respiratory symptoms, which is the key-point of this paper.
To investigate the association between RR, SpO2, and various blood test findings in respiratory diseases, authors should first show the relationship between the combined these data of all cases in the GA and GC groups and each respiratory symptoms in diseases.
And, to compare these findings in chronic and acute diseases, authors should discuss the results by comparing cases with normal pH from the GC (as the actual GC) group and cases with abnormal pH from the GA (as the actual GA) group.
Regarding the definition of acute and chronic diseases, to our knowledge, there are currently no accurate definitions of acute and chronic diseases in veterinary medicine. For example, feline rhinitis is defined as chronic rhinitis if clinical signs persist for more than 1 month (Reed, Canine and Feline Respiratory Medicine: An Update, 2020). And as your comment suggests, classification based on pH fluctuations would be one way to differentiate between acute and chronic. However, in clinical practice, even cases that are considered acute diseases such as rhinitis due to foreign body and mild canine infection respiratory diseases complex may include cases in which the pH is normal. Also, in brachycephalic obstructed airway syndrome, which is a chronic disease, the pH is very often acidic due to an increase in PCO2. Furthermore, as is the purpose of this study, there have been no studies that have conducted blood gas analysis in all acute and chronic respiratory diseases, and we believed that it was first necessary to understand the variations in these tests in respiratory diseases. Therefore, in this study, we classified based on the duration of clinical signs as in the report on rhinitis above.
Furthermore, regarding respiratory symptoms, it is difficult to compare since the amount of data is enormous and this study focused on the trends and fluctuations of each test item. Further investigation on the relationship with clinical symptoms in the future based on the present results is needed. We commented in conclusion section (revised manuscript: lines 502-504).
Plural and singular expressions should be unified in Table 1.
As you suggested, the description methods in Table 1 have been unified (revised manuscript: Supplementary Materials Table S1).
The column headings for the classification of Anatomical sites and final diagnosis are difficult to understand in Table 3.
According to your comment, column headings in Table 3 have been corrected (revised manuscript: Table 1).Reviewer 2 Report
Comments and Suggestions for Authors
Author Response
We are sending a manuscript above mentioned after revisions according to the reviewers’ comments. Revised sentences and words are highlighted in red in the manuscript. We appreciate helpful suggestions offered by reviewers. We hope the revised manuscript is acceptable for publication in MDPI Veterinary Science.
Thank you for giving me a chance for reviewing your manuscript. In this study, the relationship of RR, SpO2 or blood test results to respiratory disease in dogs were retrospectively evaluated. These findings were compared between chronic phase and acute phase. The author found some relationship between higher RR, higher CRP or lower SpO2 and lung disease, and higher WBC counts and lung or pleural diseases. The authors also found that respiratory acidosis is common in acute respiratory disease while respiratory alkalosis is more common in chronic respiratory disease. I believe that this manuscript provides clinical useful information. On the other hand, the presented data is voluminous and complicated, and difficult to understand. I strongly recommend the tables are reorganized to make them easier to read. There are some serious concerns with this paper. Firstly, the validity of the reference values used in this study is questionable. Secondly, the criteria of acid-base disorder used in this study is also questionable. The difference between alkalosis or acidosis and alkalemia or acidemia should be clarified. See specific comments for each section.
We appreciate your comment. According to your comment, we have revised the manuscript and table. Details are listed below.
Abstract
See comment below, especially in the materials & methods and results of the manuscript.
Introduction
Line 56: What is difference between ventilatory and respiratory function? I think “ventilation or oxygenation” is more appropriate.
According to your comment, this description was changed (revised manuscript: line 68).
Lines 68-70: Please include references and specific descriptions.
According to your comment, references and specific explanations have been added to this description (revised manuscript: lines 80-83).
Materials and Methods
Comment 1: I’m not sure why you have not evaluated the relationship between body temperature and respiratory disease.
As per your comment, we should have investigated the relationship between body temperature and respiratory diseases as well, but in this study, we were not able to examine records of body temperature and therefore cannot evaluate them. We commented in conclusion section (revised manuscript: lines 502-504).
Lines 97-106: I understand the limitations of retrospective study. However, accuracy of diagnosis in this study has a significant impact on the results. Please include some information on the number of veterinarians involved in the diagnosis and their experience.
As you suggested, added details about the veterinarian who made the diagnosis to this description (revised manuscript: lines 112-114).
Lines 107-119: When was the measurement (RR, SpO2, and blood tests) performed? How to count RR in this study? For 1 min or 30 sec? Please include the name of devices used in various blood tests.
As you suggested, added additional information about various tests and the name of devices used in this study to this description (revised manuscript: lines 126-128, 132-133, 136-137).
Lines 118-119, Table 2: Are you saying that the diagnosis of acid-base disorders was made based on whether all the criteria items (pH, PCO2, HCO3, BE) were met? How was the mixed acid-base imbalance (metabolic and respiratory) diagnosed? Although diagnosed as alkalosis or acidosis in your results or discussion, the diagnosis classified in the criteria (Table 2) are alkalemia and acidemia. The difference between alkalosis or acidosis and alkalemia or acidemia should be clarified. I have never seen the word like metabolic or respiratory acidemia or alkalemia. I don’t agree with the classification and analysis.
In this study, the diagnosis of acid-base disorders was based on whether all criteria items were met. Diagnostic criteria for mixed acid-base disorders have also been added to Table 2 (revised manuscript: Supplementary Materials Table S2).
There were some errors in the expressions for acidemia and alkalemia and acidosis and alkalosis. In main manuscript, acidosis and alkalosis were unified as acid-base disorders (revised manuscript: lines 140-143).
Table 1, Table 2: Table 1 and 2 should be included as supplementary materials. What are the references cited for these criteria of blood gas analysis? The pH, PCO2, and BE of vein are consistent with mean SD in reference 12. However, the HCO3 of vein and all parameters of artery are not consistent in reference 11 or 12. In particular, the reference value range of arterial PCO2 is too large. I have never seen a reference value like this. In addition, only 68% of the population (healthy dog) is within mean SD. It is appropriate as a reference value?
According to your comment, Tables 1 and 2 have been included as supplementary material (revised manuscript: Supplementary Materials Tables S1 and S2). There are still not many reports on reference values for venous blood gas analysis, and it is natural that values vary. Therefore, it is difficult to determine which report to adopt. In this study, acidosis and alkalosis were classified cause, but the main purpose of this study was to investigate the differences in the relative trends of each respiratory disease in the acute and chronic phases, without focusing on numerical values. Therefore, we referred to reference 12, which has the least variation in the reference ranges for pH and PCO2 among previous reports (reference 11, 12, and Vanova-Uhrikova et al., 2017 and Bachman et al., 2018), and some revisions have been made to the reference ranges of blood gas analysis based on reference 12 (revised manuscript: Supplementary Materials Table S2). Also, some results have been changed due to changes in reference values (revised manuscript: lines 321-323, 325-327, 333-334, Table 4).
Table 1: I think this table shows reference values of healthy dogs for various tests used in this study, not criteria.
According to your comment, this description was changed (revised manuscript: Supplementary Materials Table S1).
Results
Comment 1: How about comparing the number of dogs above or below the reference value? In each group or subgroup, the majority seemed to be within the reference values of SpO2 and blood tests except for CRP. The median values themselves don’t seem to contain clinical important values in many of the data presented in this study. It would make more sense to identify which parameters are more likely to be outlines in each disease group except for RR and CRP
As you suggested, we have added some numbers and percentages of dogs above or below the value to the main manuscript (revised manuscript: lines 217-223, 240-259, 290-301).
Lines 257-269, Table 6: As indicated earlier, I have never seen the word like metabolic or respiratory acidemia or alkalemia. The difference between alkalosis or acidosis and alkalemia or acidemia should be clarified before reaching these results.
As mentioned above, these terms have been unified with acidosis and alkalosis in main manuscript (revised manuscript: lines 140-143, Table 4).
Table 3: Why is chronic bronchitis included in the GA group?
According to your comment, some of the items in the table have been changed (revised manuscript: Table 1).
Table 4, Table 5: Too difficult and cumbersome to read. Please organize it in an easy to read format. What is the meaning of showing the reference values in Table 1 or listing all actual measured values in Table 4 and Table 5?
According to your comment, Tables 4 and 5 have been revised by deleting some items. The purpose of this study was to investigate the trends and fluctuations of each test item by disease site, so the actual measured values are listed in a table (revised manuscript: Tables 2 and 3).
Discussion
Line 319: In Table 5, the median SpO2 of dogs with pulmonary edema in GA is 97%. Are the descriptions contradictory?
There was an error in the numbers listed in the table. The figures have been changed to the corrected values (revised manuscript: Table 3).
Lines 329-331: Why didn’t you analyze PaO2 in GA? In GA, there are 139 cases of arterial blood gas analysis.
There was an error in this description, so we have corrected it. The comparison between GC and GA could not be made because GC did not include cases in which arterial blood was used. Evaluation of arterial blood was conducted within GA, and some of the results have been added to the main manuscription (revised manuscript: lines 319-320, 428-431).
Reviewer 3 Report
Comments and Suggestions for Authors
This was an interesting paper, but I have a couple of questions.
Is it known that during the SpO2 measurements was the heart rate on the pulse oximeter the same as the dogs heart rate, if the heart rate on the pulse oximeter is inaccurrate then the SpO2 value may not be correct.
When taking the SpO2 measurements how many measurements were taken, and were they averaged? Or was just one measurement taken over what time period? If one site did not give a proper reading was another site tried?
Arterial blood gases were run, but why was there no value for PaO2? I would have liked to see the correlation between the PaO2 and the SpO2. Did the arterial blood gas not do oxygen?
Author Response
We are sending a manuscript above mentioned after revisions according to the reviewers’ comments. Revised sentences and words are highlighted in red in the manuscript. We appreciate helpful suggestions offered by reviewers. We hope the revised manuscript is acceptable for publication in MDPI Veterinary Science.This was an interesting paper, but I have a couple of questions.
Is it known that during the SpO2 measurements was the heart rate on the pulse oximeter the same as the dogs heart rate, if the heart rate on the pulse oximeter is inaccurate then the SpO2 value may not be correct.
When taking the SpO2 measurements how many measurements were taken, and were they averaged? Or was just one measurement taken over what time period? If one site did not give a proper reading was another site tried?
We appreciate your comment. In this study, SpO2 was measured based on the match between the heart rate on the pulse oximeter and the dog’s heart rate in all cases. SpO2 measurements were based on single results rather than averages, and if stable measurements cannot be taken at one site, it was considered not to be accurate value and measurements were taken at another site.
Arterial blood gases were run, but why was there no value for PaO2? I would have liked to see the correlation between the PaO2 and the SpO2. Did the arterial blood gas not do oxygen?
   PaO2 values were obtained in all cases in which arterial blood gas were performed. However, because the number of cases was biased between disease sites and no significant difference was observed, and because all cases in GC were from venous blood, it was not possible to compare between GC and GA. Additionally, the correlation between PaO2and SpO2 cannot have been evaluated due to limited number and further investigated is required. We commented that in discussion and conclusion section (revised manuscript: lines 428-431, 505-507).Round 2
Reviewer 2 Report
Comments and Suggestions for Authors
I think the descriptions of acidemia or alkalemia in Table S2 are incorrect. Acidosis or alkalosis is appropriate.
Author Response
We are sending a manuscript above mentioned after revisions according to the reviewers’ comments. Revised sentences and words are highlighted in red in the manuscript. We appreciate helpful suggestions offered by reviewers. Also, in this revision, only Table S2 was revised, so the main manuscript remains unchanged.I think the descriptions of acidemia or alkalemia in Table S2 are incorrect. Acidosis or alkalosis is appropriate.
According to your comment, some notations in Table S2 have been changed (revised manuscript: Supplementary Materials Table S2).Reviewer 3 Report
Comments and Suggestions for Authors
the revisions make this a much better paper!
Author Response
We are sending a manuscript above mentioned after revisions according to the reviewers’ comments. Revised sentences and words are highlighted in red in the manuscript. We appreciate helpful suggestions offered by reviewers. Also, in this revision, only Table S2 was revised, so the main manuscript remains unchanged.
The revisions make this a much better paper!
We appreciate your comment.